# Deleterious Association of Inhalant Use on Sleep Quality during the COVID-19 Pandemic

**DOI:** 10.3390/ijerph182413203

**Published:** 2021-12-15

**Authors:** Deepti Gunge, Jordan Marganski, Ira Advani, Shreyes Boddu, Yi Jan Ella Chen, Sagar Mehta, William Merz, Ana Lucia Fuentes, Atul Malhotra, Sarah J. Banks, Laura E. Crotty Alexander

**Affiliations:** 1Department of Medicine, Pulmonary Critical Care Section, Veterans Affairs (VA) San Diego Healthcare System, La Jolla, CA 92161, USA; deeptigunge98@gmail.com (D.G.); inadvani@ucsd.edu (I.A.); yjc012@ucsd.edu (Y.J.E.C.); samehta@ucsd.edu (S.M.); wmerz@ucsd.edu (W.M.); afuentes@health.ucsd.edu (A.L.F.); amalhotra@health.ucsd.edu (A.M.); sbanks@health.ucsd.edu (S.J.B.); 2Department of Medicine, Division of Pulmonary, Critical Care and Sleep Medicine, University of California San Diego (UCSD), La Jolla, CA 92093, USA; jmargans@health.ucsd.edu (J.M.); b.shreyes@gmail.com (S.B.); 3Department of Neurosciences, Division of Pulmonary, University of California San Diego (UCSD), La Jolla, CA 92093, USA

**Keywords:** vaping, electronic cigarettes, sleep latency, marijuana, cocaine

## Abstract

The COVID-19 pandemic generated large amounts of stress across the globe. While acute stress negatively impacts health, defining exact consequences and behavioral interventions can be difficult. We hypothesized that a generalized increase in stress and anxiety caused by continuation of the global pandemic would negatively impact sleep quality and that ever users of e-cigarettes and conventional tobacco would have more profound alterations over time. Participants were recruited via social media to complete an online survey in April 2020 (n = 554). Inhalant use was assessed through the UCSD Inhalant Questionnaire and sleep quality was gauged through the Pittsburgh Sleep Quality Index (PSQI). A set of participants (n = 217) retook the survey in June 2020. Inhalant users—historical or current e-cigarette vapers, conventional tobacco smokers, and dual users—had higher PSQI scores than never smoker/never vapers, demonstrating worse sleep quality in inhalant users. Non-smoking/non-vaping subjects who retook the survey in June 2020 had improvement in their PSQI scores by paired t test, indicating better sleep quality as the pandemic continued, while inhalant users of all types had persistently high PSQI scores (poor sleep quality). These data suggest that ever users of tobacco products may be susceptible to overall diminished sleep quality in the setting of stressful life circumstances. These data also suggest that pandemic-initiated lifestyle changes may have led to improvements in sleep quality. Finally, these findings raise concerns for correlations between either past or active e-cigarette use on sleep, and thus overall health.

## 1. Introduction

While there is strong evidence linking conventional tobacco use with sleep disturbance, the effects of electronic (e)-cigarettes on sleep quality remain largely unknown. Nicotine is considered one of the causal factors in the adverse effects of tobacco on sleep quality due to its effects as a stimulant resulting in decreased sleep time and increased nighttime awakenings. Over ninety percent of e-cigarettes contain nicotine, and thus it is likely that vaping will also impact sleep quality due to the nicotine content [1,2]. In our previous study, we found that female users of both conventional tobacco and e-cigarettes, known as dual users, had worsened sleep quality and higher sleep latency (amount of time to fall asleep) than never smokers/never vapers [3]. This pilot study was replicated in a larger cohort, validating our findings that dual use is associated with increased sleep latency (*manuscript in revision*).

Although there have been several studies published over the last few years attempting to address whether e-cigarettes or dual use impact sleep health, many of the studies focused on particular age groups or did not use validated questionnaires for evaluating sleep quality. Others only looked at one type of inhalant rather than all four groups. Of the eleven articles reviewed, one only alluded to a possible relationship between e-cigarette use, oxidative stress, and adverse effects on sleep [4]. Only one study did not find a difference in sleep duration or quality in e-cigarette users; however, this study design had concerning limitations using unvalidated methods for assessing sleep [5]. The remaining nine articles had findings supportive of a relationship between e-cigarette use and decreased sleep quality including sleep duration, onset, latency or regularity [6,7,8,9,10,11,12,13,14]. In summary, although the articles all had unique limitations from study design to generalizability of findings, the vast majority of the articles concluded that e-cigarette or dual use leads to alterations in sleep that need further investigation.

Increased life stressors are commonly associated with disruptions to the sleep cycle. During the COVID-19 pandemic, many lifestyle changes were required for the global population. These included changes to work environment, childcare, social interactions, physical activity, etc. These alterations to normal routine inevitably bring about new feelings that are sometimes positive but often negative, stressful, and anxiety-provoking. Further, the pandemic itself led to fear of dying, fear of spreading the virus, and fear of losing loved ones as another layer of stress. According to epidemiologic data provided by the Centers for Disease Control (CDC) and the World Health Organization (WHO) [15,16], between the two timepoints of the surveys the globe was experiencing a trend of increasing daily COVID cases. In the same timeframe, the US was experiencing a spike in daily deaths and a plateau in daily cases of COVID-19. Because sleep is negatively impacted during times of increased stress [17,18], we hypothesized that the negative correlations between historical or active use of nicotine-based inhalants may be amplified as the COVID-19 pandemic progressed.

Here we conducted a social media-based survey during the time of COVID-19 to assess the impact of anytime use of nicotine-based inhalant use on sleep quality in the setting of global stress. We recruited participants who either in the past or present had been e-cigarette vapers, conventional tobacco users, dual users of both e-cigarettes and conventional tobacco and never smokers/never vapers to take our University of California San Diego (UCSD) Inhalant Survey online [3,19]. Our survey included detailed questions on sleep patterns and quality, at two separate time-points during the pandemic. We hypothesized that nicotine-based inhalant users would have worse sleep quality in the setting of the ongoing stress of the pandemic because they would not adapt as well as non-inhalant users.

## 2. Materials and Methods

This study was approved by the UCSD Institutional Review Board (protocol #160204). Inhalant habits involving both historical and active use were surveyed using our established UCSD Inhalant Questionnaire, while sleep quality was assessed via the Pittsburgh Sleep Quality Index (PSQI; scored from 0 to 21, with higher numbers indicating worse sleep quality). Specifically, types of inhalants and amount of vaping and smoking per day were both quantified. The years of tobacco use were included in the historical smoker questions. Participants were recruited through online advertisements posted to social media sites including Facebook, Craigslist, Reddit, and Twitter. All participants underwent informed consent prior to taking the survey. Participants (n = 554) were recruited in April 2020, and a subset (n = 217) retook the survey in June 2020. In the June retake survey, six questions were added to assess the subjects’ current level of COVID concern. Specifically, these questions included: “How concerned about COVID are you right now? How much has COVID disrupted your life? Has the COVID-19 pandemic caused you anxiety? Since the pandemic began, how has your work-related stress changed? Have you contracted COVID? Have any family members, friends, or loved ones contracted COVID?” All participants were entered into a weekly lottery for a USD 250 Amazon gift card.

Data were analyzed with GraphPad Prism (version 9.0.2, GraphPad software, San Diego, CA, USA). Descriptive analyses were performed for demographics and sleep quality outcome scores, with outcomes being summarized by both inhalant use groups and overall. Differences between inhalant users and non-inhalant users were analyzed by Welch’s two-tailed *t*-test, and longitudinal changes were analyzed with paired *t*-tests. Differences across e-cigarette users, conventional smokers, dual users and non-inhalant users, both before and during COVID, were analyzed with 2-way ANOVA with Sidak for multiple comparisons.

## 3. Results

### 3.1. Demographics

A total of 217 participants completed the online survey. One hundred and forty-nine were never smoker/never vapers, 39 were conventional tobacco smokers, 10 were e-cigarette or vaping device users, and 19 participants were dual users. The majority of participants identified as women (74%), with 25% men, and 1% non-binary. The majority of subjects (n = 139) self-identified as Caucasian. The ages of participants ranged from 18 to 75 years, with a considerable fraction of participants grouped between the ages of 21 to 30 (39%) (Table 1). Eighty-five percent of participants were located across the US with the remaining dispersed internationally including Australia, the United Kingdom, Canada, Israel, and Germany (Figure 1).

### 3.2. Association between Inhalant Use and Sleep Quality

In the setting of the global pandemic (June 2020), past or present inhalant users overall had higher PSQI scores, indicating worse sleep quality (6.221), relative to never smoker/never vapers (5.248, *p* = 0.012; Figure 2A). These data may suggest that any lifetime use of a nicotine containing inhalant (conventional tobacco, e-cigarettes, or both) is adversely associated with sleep quality, and that this association may be magnified in the presence of a continued stressor (in this study, the identified stressor is uncertainty and anxiety associated with an unresolved global pandemic). Even when controlling for age, gender and presence of any lung disease, inhalant use significantly was associated with a higher PSQI score. Using multivariate regression, inhalant use, age and lung disease were all independent predictors of the PSQI score. Advancing age predicted lower PSQI scores, with the parameter estimate for age being −0.030 per year (CI −0.008 to −0.053; *p* = 0.0087). PSQI score was also higher in people with lung disease, with the parameter estimate being 1.040 if present (CI 0.029 to 2.051; *p* = 0.0439). Finally, inhalant use predicted higher PSQI score, with a parameter estimate of 0.810 (CI 0.087 to 1.532; *p* = 0.0283).

### 3.3. Change in Sleep Quality during the Pandemic

When survey participants retook our survey two months later in June 2020, sleep quality for the group as a whole improved, as shown by lower PSQI scores (5.595), relative to the original survey in April 2020 (7.990; mean of differences (retake–original) = −2.395, SD ± 3.256, *p* < 0.0001; Figure 2B). This finding suggests that participants had improvement in sleep quality as the COVID pandemic progressed. However, when broken down by inhalant type, never smoker/never vapers were the only group that had a significant improvement in sleep quality (*p* < 0.0001; Figure 2C and Table 2). This suggests that non-inhalant users were better able to adapt during the pandemic leading to improved sleep quality while nicotine inhalant users were less able to adapt and did not have improvements in sleep quality.

To assess what factors might be contributing to sleep quality during the stress of the pandemic, six questions were included in the retake survey in June to assess for the subjects’ mindsets related to COVID. The average response across inhalant groups on a scale of 1–10, with 10 representing survey respondents being extremely concerned about COVID-19, was 6.28 (Figure 3A). The average response across inhalant groups with 10 representing survey respondents having their lives extremely disrupted by COVID-19, was 7.24 (Figure 3A). The largest portion of e-cigarette users felt that their work-related stress increased a little where the largest portion of dual users felt that their work stress increased a lot (Figure 3B). Eighty-four percent of survey participants reported not having contracted COVID-19 at the time of the survey (Figure 3C). Twenty-two percent of participants reported that a friend, family member, or loved one had contracted COVID-19 and tested positive (Figure 3D). The pandemic caused 61.8% of our participants “a little” anxiety and 31.3% “a lot” of anxiety (Figure 3E).

## 4. Discussion

To our knowledge, these data are the first to find an adverse association of lifetime e-cigarette use on overall sleep quality. While we have run larger survey-based studies and have found adverse effects of dual use of e-cigarettes and conventional tobacco on certain aspects of sleep [3], the findings presented here suggest that inhalant users may be susceptible to overall diminished sleep quality in the setting of continued stressful life circumstances. In addition, our finding that non-inhalant using subjects had improvements in sleep quality as the pandemic went on, while none of the inhalant groups had similar improvements, lends support to the idea that nicotine use via any inhalant device may lead to poor sleep quality and a lack of adaptability in the time of stress.

With the vast majority of the population either working from home during the pandemic or not working at all, we postulated that the improvement in PSQI scores in never smoker/never vapers from April to June 2020 may have been driven by lifestyle changes. Contributing factors to the improved PSQI scores could include extended sleep duration and decreased variability of bedtimes [19]. The specific etiologies of the changes in sleep duration are an area of research that will require further investigation but could include decreased commute time, as well as decreased time spent in preparation for working outside of the home (i.e., grooming time, meal preparation).

This study has several limitations. The sample size was small, with small numbers of e-cigarette and dual users in particular. Larger studies are needed to confirm these findings as well as studying broader populations. Our cohort had significant age differences between the e-cigarette group and the other inhalant groups; however, multivariate analysis was able to define the impact of age as well as inhalant use on PSQI scores. As this survey-based study relied on participants’ self-reporting, there may have been inaccuracy in responses. However, we anticipate such misclassification to be random and thus should bias towards the null hypothesis. A baseline, pre-pandemic PSQI score was not collected. Nonetheless, this study having multiple time points of data collection during the pandemic opens the door for eventual comparison to post-pandemic PSQI scores. Another limitation of this study is that it is only able to infer correlation between inhalant use and sleep quality, not causation. As such, we are only able to discuss the associations between inhalant use and sleep quality as the pandemic progressed, not impact or effects. The study assessed the relationship between the COVID pandemic and the participants lifestyle with a series of six questions that were added to the retake survey. Ideally, the study would have included more questions to evaluate the degree of limitations and emotional responses to the stressor of the global pandemic in more depth. However, in the interest of being able to recruit and retain participants, the number of questions within our survey were minimized. In addition, because of the multiple locations of study participants throughout the US and worldwide, it was not possible to define characteristics of the pandemic, including severity and evolution, at each location. Finally, as with any study that uses social media as the recruitment method, there is potential for sampling and participation bias.

## 5. Conclusions

In this study, we found differences in sleep quality associated with history of inhalant use. As global events including the COVID-19 pandemic continue to occur, it is important for the general public to be informed about the possible adverse associations that nicotine-based inhalants could be having on sleep quality. As poor sleep quality is known to be detrimental to general health and wellbeing, we view measures to improve sleep and immune function as important strategies for disease prevention.

## Figures and Tables

**Figure 1 ijerph-18-13203-f001:**
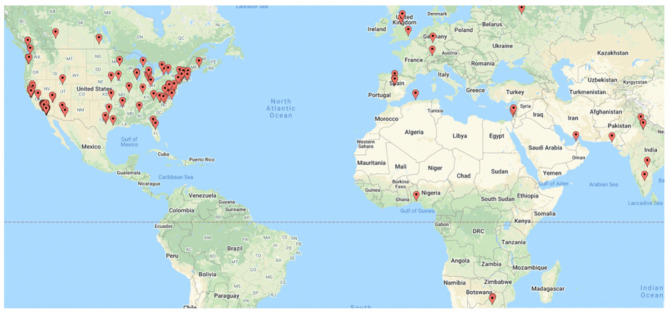
Locations of survey participants.

**Figure 2 ijerph-18-13203-f002:**
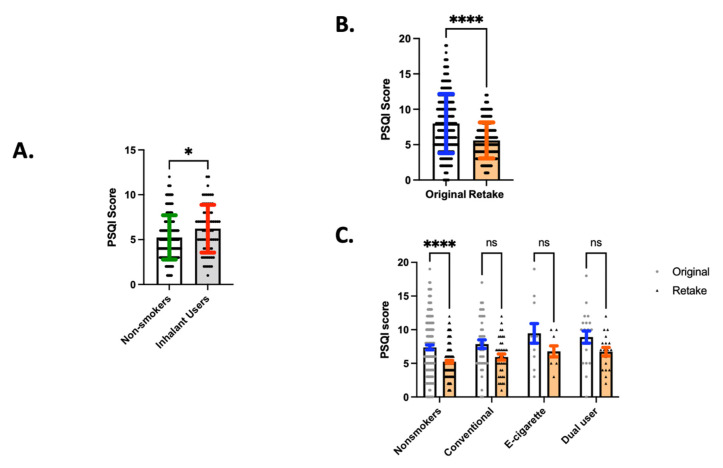
(**A**) Inhalant use associated with poorer sleep quality. Nonsmokers reported a lower PSQI score in the setting of the COVID pandemic relative to inhalant users, which comprises of e-cigarette users, conventional tobacco users, and dual users. (**B**) PSQI scores in the original survey, administered in April 2020, were higher than PSQI scores in the retake survey, administered in June 2020. (**C**) Nonsmokers had higher PSQI scores in June 2020 than in April 2020, whereas other inhalant groups had no significant change in PSQI scores during this time. * *p* < 0.05, **** *p*< 0.0001.

**Figure 3 ijerph-18-13203-f003:**
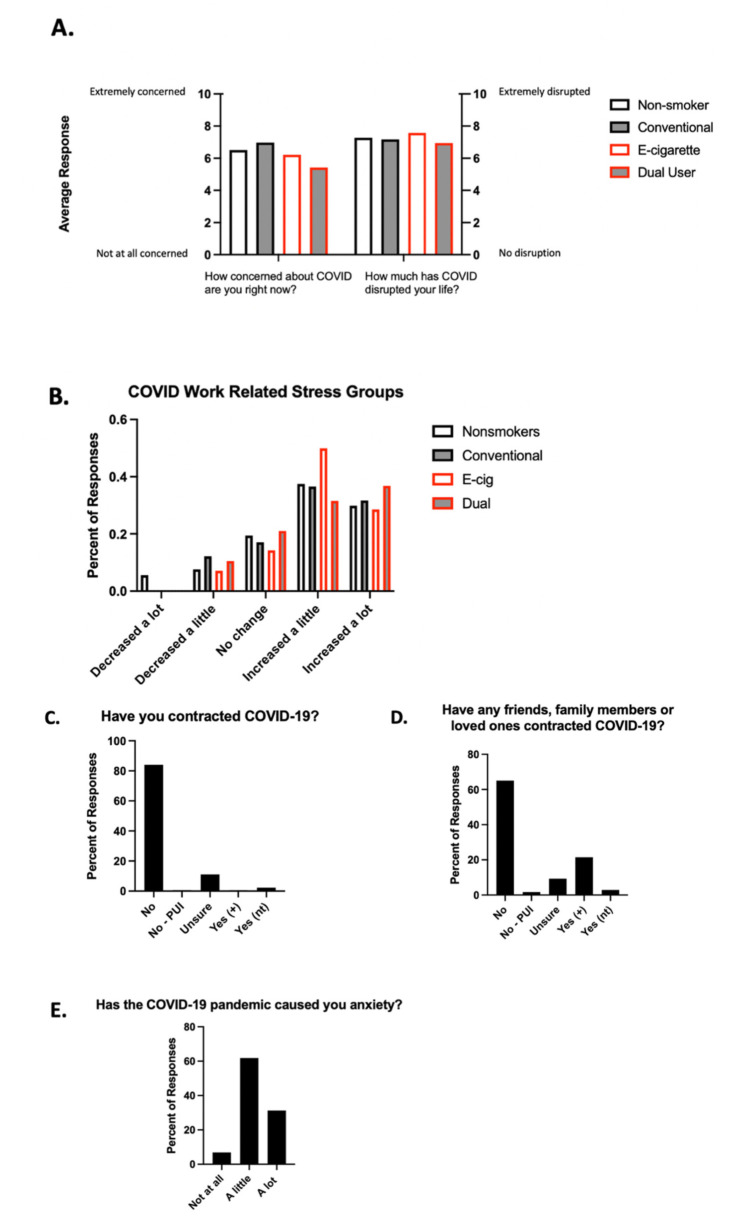
Non-inhalant and inhalant users experienced anxiety and work-related stress during the COVID pandemic. (**A**) Responses to “How concerned about COVID are you right now?” and “How much has COVID disrupted your life?” Answers were submitted free response to these questions on a scale from 1–10. Average responses for the entire group of participants are presented. (**B**) Assessment of work-related stress caused by the pandemic across inhalant groups. Participants responded by selecting one of the multiple-choice options: decreased a lot, decreased a little, no change, increased a little, increased a lot. (**C**,**D**) Rates of COVID-19 in survey responders and their loved ones, expressed as percent of total responses. Answer options were the following: no; no but was a person under investigation (No–PUI); unsure; yes and tested positive (Yes (+)); and yes but was not tested (Yes (nt)). (**E**) Assessment of anxiety caused by the COVID-19 pandemic, expressed as percent of total responses. Answer options included not at all, a little, and a lot.

**Table 1 ijerph-18-13203-t001:** Survey Participant Demographics.

		Frequency	Percent
**Group Category**	Non-smoker	149	68.7
	Conventional tobacco	39	18.0
	E-cigarette	10	4.6
	Dual User	19	8.8
	Total	217	100
**Gender**	Male	54	24.9
	Female	161	74.2
	Non-binary	2	0.9
	Total	217	100
**Race**	Caucasian	139	64.1
	African American	4	1.8
	Asian	64	29.5
	Other	10	37.0
	Total	217	100
**Ethnicity**	Hispanic	21	9.7
	Non-Hispanic	192	88.5
	Missing	4	1.8
	Total	217	100
**Age**	18–20	19	8.8
	21–30	84	38.7
	31–40	38	17.5
	41–50	28	12.9
	51–60	24	11.1
	61+	24	11.1
	Total	217	100

**Table 2 ijerph-18-13203-t002:** PSQI Scored by Inhalant Group Comparing Original to Retake.

PSQI					
	Original	Retake	Predicted Mean Diff.	95% CI of Diff.	*p*–Value
Non-smoker/ Non-vapers	7.360	5.236	2.124	1.094 to 3.153	<0.0001
Conventional	7.850	5.947	1.903	−0.1104 to 3.916	0.0720
E-cigarette	9.455	6.778	2.677	−1.317 to 6.671	0.3274
Dual Users	8.900	6.722	2.178	−0.7093 to 5.065	0.2188

## Data Availability

Data will be made freely available to any interested parties.

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
