# Peer review of "Deleterious Association of Inhalant Use on Sleep Quality during the COVID-19 Pandemic"

_ijerph, 2021, doi:10.3390/ijerph182413203_

Round 1

Reviewer 1 Report

- The introduction is extremely brief and does not sufficiently address the rationale of the study and how this study links to past studies. The authors should at least discuss how the two variables they tested relate to one another (i.e., inhalants use and sleep quality). Also, the study seems oversimplified and repetitive of past research if no mediators/moderators are included in the study. - Another main concern is that the authors "hypothesized that the adverse effects of historical or active use of nicotine-based inhalants may be amplified in the setting of the COVID-19 pandemic." But this is NOT what they are able to address. They did not compare before and after COVID-19. Any effects found could be due to a lot of environmental factors, but not necessarily COVID. Implying that their effects were solely due to COVID is misleading. The authors include NO COVID related variables. If they want to demonstrate that the effect is related to COVID, then they at least have to include participants' perception of COVID. The two survey timepoints were April 2020 and June 2020. Both are during the pandemic, so nothing can be inferred about COVID based on comparing their scores in these two timepoints. - The measures were not sufficiently discussed. How was the PSQI calculated? How was inhalant habits being asked? Did the survey ask about frequency or amount of use or years of use? Measuring differently could mean very different thing.

Author Response

1. The introduction is extremely brief and does not sufficiently address the rationale of the study and how this study links to past studies. The authors should at least discuss how the two variables they tested relate to one another (i.e., inhalants use and sleep quality).

Author’s response: We have expanded the introduction to include further information to explain the connections being drawn between nicotine use, sleep quality stress, and the global pandemic. Page 1 and 2, “It is known that conventional tobacco disrupts sleep quality. Nicotine is considered one of the causal factors in the adverse effects of tobacco on sleep quality due to its effects as a stimulant resulting in decreased sleep time and increased nighttime awakenings. Over ninety percent of e-cigarettes contain nicotine, and thus it is likely that vaping will also impact sleep quality due to the nicotine content. Increased life stressors are commonly associated with disruptions to the sleep cycle. During the COVID-19 pandemic, many lifestyle changes were required for the global population. These included changes to work environment, childcare, social interactions, physical activity, etc. These alterations to normal routine inevitably bring about new feelings that are sometimes positive but often negative, stressful, and anxiety-provoking. Further, the pandemic itself led to fear of dying, fear of spreading the virus, and fear of loved ones dying as another layer of stress. Because sleep is negatively impacted during times of increased stress [2], we hypothesized that the adverse effects of historical or active use of nicotine-based inhalants may be amplified as the COVID-19 pandemic progressed.”

2. Also, the study seems oversimplified and repetitive of past research if no mediators/moderators are included in the study.

Author’s response: Clarifications were added to the end of the introduction section to further delineate where our study fits into the existing related literature. Page 2, “Caviness et al established an association between conventional tobacco cigarette use and decreased sleep quality using the PSQI questionnaire. Riehm et al found associations between e-cigarette use, dual use, and sleep-related complaints however, they did not use a validated sleep quality assessment tool such as the PSQI for assessment. They asked a single unvalidated question: When was the last time that you had significant problems with sleep trouble, such as bad dreams, sleeping restlessly, or falling asleep during the day? And they scored respondents as having sleep-related complaints if they responded positively for complaints in the last year. Our study expands upon both of these previous works by using the validated PSQI tool to evaluate sleep quality in subjects who use e-cigarettes, cigarettes, neither, or both.”

3. Another main concern is that the authors "hypothesized that the adverse effects of historical or active use of nicotine-based inhalants may be amplified in the setting of the COVID-19 pandemic." But this is NOT what they are able to address. They did not compare before and after COVID-19. Any effects found could be due to a lot of environmental factors, but not necessarily COVID. Implying that their effects were solely due to COVID is misleading. The authors include NO COVID related variables. If they want to demonstrate that the effect is related to COVID, then they at least have to include participants' perception of COVID. The two survey timepoints were April 2020 and June 2020. Both are during the pandemic, so nothing can be inferred about COVID based on comparing their scores in these two timepoints

Author’s response: We agree with the reviewer and the ideal study would have been following a cohort established before the pandemic to definitively find out if their sleep quality changed in the setting of the pandemic. We have clarified in the introduction that the study was run during the pandemic without an earlier timepoint. However, we did include six COVID-specific questions at the second timepoint. Specifically, some of these questions were, “How concerned about COVID are you right now? How concerned about COVID were you one month ago? Has the COVID-19 pandemic caused you anxiety?” We added these questions to try to assess whether inhalant users were experiencing greater COVID-related stress, which we thought might be due to their inhalant use or underlying behavioral factors. We have now added this data to our results section and feel that it has enriched the manuscript. Our hypothesis is more accurately stated as: as the pandemic (stressor) progressed and continued, there would be worse sleep quality among historical or current inhalant users because these subjects would not adapt to the stress of the pandemic as adeptly as non-vapers.

4. The measures were not sufficiently discussed. How was the PSQI calculated? How was inhalant habits being asked? Did the survey ask about frequency or amount of use or years of use? Measuring differently could mean very different thing.

Author’s Response: Page 2, “Inhalant habits involving both historical and active use were surveyed. Types of inhalants and amount per day were both quantified. The years of use were included in the historical smokers questioning. PSQI is a questionnaire that is scored from 0 to 21 with higher numbers indicating worse sleep quality.”

Reviewer 2 Report

The study titled "Deleterious impact of inhalant use on sleep quality during the COVID-19 pandemic" is a good study but not very clear in terms of details of statistical analysis. Lots of additional analyses can be done to improve the quality of paper. 

Author Response

1. The study titled "Deleterious impact of inhalant use on sleep quality during the COVID-19 pandemic" is a good study but not very clear in terms of details of statistical analysis. Lots of additional analyses can be done to improve the quality of paper.

Author’s Response: We agree with the reviewer that the manuscript could benefit from more specificity in regard to the data collection and analysis. To address this concern, more detailed information was included in the materials and methods and results sections. Page 2, “Inhalant habits involving both historical and active use were surveyed. Types of inhalants and amount per day were both quantified. The years of use were included in the historical smokers questioning. PSQI is a questionnaire that is scored from 0 to 21 with higher numbers indicating worse sleep quality.’ ‘Participants (n=554) were recruited in April 2020, and a subset (n=216) retook the survey in June 2020. In the June retake survey, six questions were added to assess the subjects’ current COVID concern. Specifically, some of these questions were, “How concerned about COVID are you right now? How concerned about COVID were you one month ago? Has the COVID-19 pandemic caused you anxiety?”’

Page 3, “When survey participants retook the survey in June 2020, sleep quality for the group as a whole improved, as shown by lower PSQI scores (6.100, SD = 2.973, p < 0.0001), relative to the original survey in April 2020 (7.918, SD = 3.493; Figure 1B). This finding suggests that participants had improvement in sleep quality as the COVID pandemic progressed. However, when broken down by inhalant type, never-smoker/never-vapers were the only group that had a significant improvement in sleep quality (p < 0.001; Figure 1C). This suggests that non-inhalant users were better able to adapt during the pandemic leading to increased sleep quality while nicotine inhalant users were less able to adapt and did not have improvements in sleep quality. To assess what factors might be contributing to sleep quality during the stress of the pandemic, six questions were added to the retake survey in June to assess for the subjects’ mindsets related to COVID. The average response across inhalant groups on a scale of 1-10, with 10 representing survey respondents being extremely concerned about COVID-19, was 6.28 (Figure 3A). The average response across inhalant groups with 10 representing survey respondents having their lives extremely disrupted by COVID-19, was 7.24 (Figure 3A). The largest portion of e-cigarette users felt that their work-related stress increased a little where the largest portion of dual users felt that their work stress increased a lot (Figure 3B). Eighty-four percent of survey participants reported not contracting COVID-19 at the time of survey collection (Figure 3C). 21.5% of participants reported that a friend, family member, or loved one had contracted COVID-19 and tested positive (Figure 3D). The pandemic caused 61.8% of our participants at least “a little” anxiety (Figure 3E).”

Reviewer 3 Report

The study is of interest in the context of the current COVID-19 pandemic

However, it is hampered by the lack of relevant information on the pandemic, both in terms of severity and evolution, and on life style changes of the participants in relation with the pandemic

The following information is missing

  1. Location of the study, San Diego area or larger area (participants recruited through online advertisements posted to social media sites)
  2. Duration of the pandemic prior to the April-June 2020 survey in the study area
  3. Severity of the pandemic at the time of the two surveys. Number of  COVID-19 new cases per 100.000 inhabitants, number of hospitalized patients
  4. Improvement or worsening of the pandemic at the time of the study
  5. Degree of limitation imposed to the subjects, total or partial confinement
  6. Life style changes: working at home / working at the office?  Advantage or disadvantage.

Time elapsed between the original survey (April 2020) and the retake survey (June 2020). 1 month? Very short. 2 months? Short. 3 months? Acceptable

Table 1. Demographics by Group Category well detailed

Figure1. Results not easily readable, specially in the case of figure C. A table would be welcome

Improvement in sleep quality in never-smoker/never-vaper subjects, between the two surveys, cannot be explained without a precise  knowledge of the evolution of the pandemic during this period and without a self-assessment of the life-change

Author Response

The study is of interest in the context of the current COVID-19 pandemic

However, it is hampered by the lack of relevant information on the pandemic, both in terms of severity and evolution, and on life style changes of the participants in relation with the pandemic

The following information is missing

  1. Location of the study, San Diego area or larger area
  2. Duration of the pandemic prior to the April-June 2020 survey in the study area
  3. Severity of the pandemic at the time of the two surveys. Number of COVID-19 new cases per 100.000 inhabitants, number of hospitalized patients
  4. Improvement or worsening of the pandemic at the time of the study

1-4.Author’s Response:

We used the IP addresses from our survey to geolocate where our participants had been when they took the survey. As recruitment of this project occurred via social medias, the reach of the study was able to extend further than our local area. Eighty-five percent of our participants were located across the US with the remaining located internationally including Australia, United Kingdom, Canada, Israel, and Germany. A map of the location of our survey participants has been added to the figure section of our manuscript. We have added the following epidemiologic data from the CDC to the manuscript: Page 2, “According to epidemiologic data provided by the CDC and the WHO, between the two timepoints of the surveys the globe was experiencing a trend of increase in daily COVID cases. In the same timeframe the US was experiencing a spike in daily deaths and a plateau in daily cases of COVID-19.”  

  1. Degree of limitation imposed to the subjects, total or partial confinement
  2. Life style changes: working at home / working at the office?  Advantage or disadvantage. Improvement in sleep quality in never-smoker/never-vaper subjects, between the two surveys, cannot be explained without a precise knowledge of the evolution of the pandemic during this period and without a self-assessment of the life-change.

5-6.Author’s Response:

In order to get adequate numbers of responses and completion rates, we designed the survey to be brief. Thus, although we agree with the reviewer that these are fascinating questions, we were unable to include all of them because the survey would have been too long and our participant number would have decreased. However, we did include six COVID-specific questions in the June retake survey. Specifically, some of these questions were, “How much has COVID disrupted your life? Has the COVID-19 pandemic caused you anxiety? Since the pandemic began, has your work-related stress changed?” We added these questions to try to assess whether inhalant users were experiencing greater COVID-related stress, which we thought might be due to their inhalant use and/or underlying behavioral factors. We have now added the data from these COVID-related questions to our results section and feel that it has enriched the manuscript. We have also added this topic to the discussion section of our manuscript as one of the limitations. Page 6, “The study assessed the impact of COVID on the participants lifestyle with a series of six questions that were added to the retake survey. Ideally, the study would have included more questions to evaluate the degree of limitations and emotional responses to the stressor of the global pandemic in more depth. However, in the interest of being able to recruit enough participants, our team felt it best that the survey duration not be extended excessively.”

  1. Time elapsed between the original survey (April 2020) and the retake survey (June 2020). 1 month? Very short. 2 months? Short. 3 months? Acceptable. Author’s Response: The interval between surveys was designed to be 2 months. Eight weeks after the participant took the original survey, the retake was sent out to that participant. The study was designed this way to obtain the most retake survey respondents by keeping it relevant and fresh in the subjects’ minds.

8. Table 1. Demographics by Group Category well detailed

  1. Results not easily readable, especially in the case of figure C. A table would be welcome. Author’s Response: Color was added to error bars in Figure 1 to aid readability. The figure itself was reorganized with Figure 1B and 1C moved to the same side with the same color format to show that 1C is a breakdown of 1B into its different inhalant groups. P values were added to the graph in Figure 1C. Table 2 was constructed to show these P values and confidence intervals more clearly.

Round 2

Reviewer 1 Report

The authors made some improvements. However, there are still some remaining issues that need to be addressed:

  • Correlation is not causation. The authors kept using causal words, e.g., "impacts" "effects" "affect", etc. to refer to correlational findings. I strongly recommend that the authors remove any reference to causation and make it very clear in the limitation that this study only infers correlation, not causation. e.g., figure 1 is strongly misleading when the authors said "inhalant use impacts sleep quality", "time of pandemic impacts sleep quality", and "inhalant use... impacts sleep quality". I highly recommend that the authors look up when one can make causal arguments. This study is in no way inferring causation. 
  • The authors included some covid questions but they did not look at how it plays a role in the relationship they hypothesized. These variables should be examined as a mediator according to what the authors suggested. 
  • The authors said "To our knowledge, these data are the first to find an adverse association of lifetime e-cigarette use on overall sleep quality." - a simple google scholar search already resulted in a lot of studies that have demonstrated this link.  
  • Even after the authors' revision, I maintain that I do not find this study to contribute any scientific knowledge in the relationship between smoking e-cigarette and sleep quality. The analyses are not appropriately conducted to shed light to what the authors claimed, "hat the adverse effects of historical or active use of nicotine-based inhalants may be amplified as the COVID-19 pandemic progressed"

Author Response

We appreciate the reviewer’s time and effort towards reviewing our manuscript. We have addressed their comments and concerns in a point-by-point manner below. We believe this has yielded a more accurate and detailed manuscript. Thank you for the opportunity to resubmit this work.

1. Reviewer Comment: Correlation is not causation. The authors kept using causal words, e.g., "impacts" "effects" "affect", etc. to refer to correlational findings. I strongly recommend that the authors remove any reference to causation and make it very clear in the limitation that this study only infers correlation, not causation. e.g., figure 1 is strongly misleading when the authors said "inhalant use impacts sleep quality", "time of pandemic impacts sleep quality", and "inhalant use... impacts sleep quality". I highly recommend that the authors look up when one can make causal arguments. This study is in no way inferring causation. 

  1. Author’s Response: We have removed the terms “affect” and “impact” from our manuscript. The word, “effects” now only is used four times in our manuscript and each time it is referring to another study/finding or piece of information, not our own current study. We have also drawn attention to the fact that our study only infers correlation, not causation, by adding this as a limitation in our discussion section. The following quotations are where changes were made.

Figure titles and captions have been changed to say “associated with” rather than “impacted” in Figure 1 (page 4) and Figure 3 (page 6 and 7). (Still need to change the titles).

Page 1: The title of the manuscript was changed to, “Deleterious association of inhalant use on sleep quality during the COVID-19 pandemic.”

Page 1: “Finally, these findings raise concerns for correlations between either past or active e-cigarette use on sleep, and thus overall health.”

Page 2: We changed the wording of our hypothesis to reflect correlation rather than causality. “Because sleep is negatively impacted during times of increased stress [6, 7], we hypothesized that the negative correlations between historical or active use of nicotine-based inhalants may be amplified as the COVID-19 pandemic progressed.”

Page 3: “These data may suggest that any lifetime use of a nicotine containing inhalant (conventional tobacco, e-cigarettes, or both) is adversely associated with sleep quality, and that this association may be magnified in the presence of a continued stressor (in this study, the identified stressor is uncertainty and anxiety associated with an unresolved pandemic).”

Page 8: “As global events including the COVID-19 pandemic continue to occur, it is important for the general public to be informed about possible adverse associations they could be having with sleep quality.”

We have added this topic to the limitation section of our manuscript. Page 8: “Another limitation of this study is that it is only able to infer correlation between inhalant use and sleep quality, not causation. As such, we are only able to discuss the associations between inhalant use and sleep quality as the pandemic progressed, not impact or effects.”

2. Reviewer Comment: The authors included some covid questions but they did not look at how it plays a role in the relationship they hypothesized. These variables should be examined as a mediator according to what the authors suggested. 

  1. Author’s Response: As we included in the last response to reviewers, these questions specific for COVID were added to the retake survey to try to determine whether the sleep quality changes could be driven by COVID lifestyle changes. We understand that these questions were not able to establish clear causality, but we do believe that they give a little bit of insight as to what might be driving the improvement in sleep quality in non-inhalant users versus no improvement or worsening in the inhalant users.

Although the impact of COVID on our findings is of major interest, we cannot draw major conclusions as all data were collected during the pandemic. Nonetheless, we did see improvements in sleep quality as the pandemic went on, primarily in non-smokers/non-vapers rather than in participants who reported smoking and vaping. Although these findings are of interest, we lack the ability to draw causal inferences based on our study design.

3. Reviewer Comment: The authors said "To our knowledge, these data are the first to find an adverse association of lifetime e-cigarette use on overall sleep quality." - a simple google scholar search already resulted in a lot of studies that have demonstrated this link.  

  1. Author’s Response: We conducted a literature review to address this concern. We examined the top 12 related research articles found when searching for “e-cigarette” and “sleep quality” on both PubMed and Google Scholar. Below we have included each paper with some of its limitations and differences in design from ours. Specifically, ours is unique in that it looks at all four inhalant groups, uses a validated questionnaire (PSQI), and studies adults >18 years old. In addition, we also found that ours was distinct from these publications in that our participants are predominantly located in the US, we used social media for recruitment, and we had a relatively high percentage of Asian participants. We understand that our study also has its limitations. However, we believe we have addressed these limitations clearly within the discussion. We are hopeful that over time with multiple studies, a clearer picture of the impact of e-cigarette use and dual use on sleep health may come into focus.

Survey 3R Literature Review

Article 1:

Tobore T. O. (2019). On the potential harmful effects of E-Cigarettes (EC) on the developing brain: The relationship between vaping-induced oxidative stress and adolescent/young adults social maladjustment. 

This article focused on the relationship between traditional cigarettes (TC), e-cigarettes (EC) and oxidative stress. On page 204 (Section 2.4) they comment on TC’s association with sleep disruption but only hypothesize about the relationship between EC and sleep disruption with oxidative stress as a mediator. This article did not directly assess sleep quality in any way. The authors simply inferred a possible effect due to e-cigarette vaping.

Article 2:

Zvolensky, M. J., D'Souza, J., Garey, L., Alfano, C. A., Mayorga, N. A., Peraza, N., & Gallagher, M. W. (2020). Subjective sleep quality and electronic cigarette dependence, perceived risks of use, and perceptions about quitting electronic cigarettes.

This study examined exclusively e-cigarette users where our study used the four groups; conventional, e-cigarette, dual, and neither. This is important because the majority of e-cigarette users are actually dual users and thus this is the most important group to study for the highest amount of relevance to public health. This makes ours a more comprehensive assessment of the impact of inhalants on sleep quality. Additionally, the demographics of our population differed. Our study population had 27% of participants that identified as Asian where this study had 3%, thus making our findings potentially more applicable to this group. They used Qualtrics for recruitment where we used social media making our populations different but likely both relevant.

Article 3:

So, C. J., Meers, J. M., Alfano, C. A., Garey, L., & Zvolensky, M. J. (2021). Main and Interactive Effects of Nicotine Product Type on Sleep Health Among Dual Combustible and E-Cigarette Users.

This study evaluated dual users only. It did not look at e-cigarette or cigarette smokers alone. The demographics between our two studies were different. 27% of our participants identified as Asian where only 2.2% of their participants did, making our study potentially more applicable to this group. They did not use social media as their recruitment strategy, making our populations inherently different. The author’s use of the term smoking with relation to e-cigarette use was highly concerning, as these devices do not generate smoke, they generate aerosols. This also leads to concerns about the quantification and assessment of e-cigarette use. These authors found certain aspects of sleep to be impacted by different use patterns of both combustible products and vaping devices. This study actually highlights the importance of having e-cigarette user only and conventional tobacco only groups as well as dual user groups to most accurately pinpoint specific effects of each as well as interactions of the two leading to more severe effects on sleep health.

Article 4:

Kang, S. G., & Bae, S. M. (2021). The Effect of Cigarette Use and Dual-Use on Depression and Sleep Quality. 

This article uses PSQI and the four inhalant groups of conventional tobacco, e-cigarette, non-smoker and dual user. However, it is performed using data collected using the Korean Community Health Survey by the Korean CDC. Our study participants are located primarily in the US. Kang & Bae found that among the smoking groups, the dual-user group had the worst sleep quality. The findings from this study, run in Korea, are complementary to the data we found in our study in the US.

Article 5:

Brett, E. I., Miller, M. B., Leavens, E., Lopez, S. V., Wagener, T. L., & Leffingwell, T. R. (2020). Electronic cigarette use and sleep health in young adults.

The population of this study is exclusively college students who were recruited from a single institution and self-selected into the study. Our study recruited via social media and throughout the US with some international participation as well. Brett et al also excluded dual users of conventional tobacco and e-cigarettes to avoid confounding results. We included dual users in our study design. Thus, our study has a broader representation of the US population as well as a broader assessment of inhalant use patterns. However, their findings that current combustible and e-cigarette users reported significantly more sleep difficulties than never users, and users of e-cigarettes reported greater use of sleep medication than combustible cigarette users are of interest because the greater use of sleep medications does suggest longer sleep latency/insomnia.

Article 6:

Riehm, K. E., Rojo-Wissar, D. M., Feder, K. A., Mojtabai, R., Spira, A. P., Thrul, J., & Crum, R. M. (2019). E-cigarette use and sleep-related complaints among youth.

This study’s population is exclusively adolescents. We studied adults >18 years old. They also did not use a validated questionnaire to evaluate sleep-related complaints. Rather, they asked a single sleep-related question; “When was the last time that you had significant problems with sleep trouble, such as bad dreams, sleeping restlessly, or falling asleep during the day?” Sleep-related complaints were considered present if the respondent selected either “in the past month” or “two to twelve months” from the response options, which yielded a dichotomous variable of past-year sleep-related complaints versus no past-year sleep-related complaints.” Our study uses the validated PSQI, which contains 20 questions that allows for detailed assessment of a wide variety of factors that contribute to sleep disruption. Thus, our study both assesses a different population (all adults vs adolescents) and utilized a validated tool to more robustly assess for impact of inhalant use on sleep quality.

Article 7:

Dunbar, M. S., Tucker, J. S., Ewing, B. A., Pedersen, E. R., Miles, J. N., Shih, R. A., & D'Amico, E. J. (2017). Frequency of E-cigarette Use, Health Status, and Risk and Protective Health Behaviors in Adolescents. 

This study population is entirely adolescents. We studied adults >18 years old. Sleep quality and duration were assessed using three questions. No validated sleep questionnaire, such as the PSQI, was used. Dunbar et al found that user groups were similar on physical health and engagement in protective health behaviors (physical activity, sleep duration/quality, healthy diet). However, because they did not assess it in a robust, validated questionnaire manner, these results are questionable.

Article 8:

Kianersi, S., Zhang, Y., Rosenberg, M., & Macy, J. T. (2021). Association between e-cigarette use and sleep deprivation in U.S. Young adults: Results from the 2017 and 2018 Behavioral Risk Factor Surveillance System.

This study collected data from a nationwide telephone-based survey. Ours used social media to recruit participants. This study also only looks at people ages18-24 where ours is everyone >18. The authors evaluated for sleep deprivation (<7 hours of sleep per 24 hour period) and used only a single question, “On average, how many hours of sleep do you get in a 24-hour period?” Thus, this study did not assess sleep quality, it specifically evaluated sleep deprivation.

Article 9:

Merianos, A. L., Jandarov, R. A., Choi, K., Fiser, K. A., & Mahabee-Gittens, E. M. (2021). Combustible and electronic cigarette use and insufficient sleep among U.S. high school students.

This study uses exclusively high school students. They asked a single question to assess sufficient levels of sleep. Students reported the number of hours of sleep they typically get on an average school night with >8h/night being deemed sufficient. In contrast to this, our study used the validated PSQI questionnaire to assess sleep quality as well as all adults >18.

Article 10:

Abafalvi, L., Pénzes, M., Urbán, R. et al. Perceived health effects of vaping among Hungarian adult e-cigarette-only and dual users: a cross-sectional internet survey. 

This study population is entirely Hungarian where ours is predominantly US participants. They assessed changes in physiological functions, one of which was sleep. Respondents were asked if they experienced worsening, no change or improvement of each listed health condition since they initiated e-cigarette use. They did not use a validated questionnaire to assess sleep quality as our study did. While the authors found that dual users (17.6%) were significantly more likely to report AEs of vaping than e-cigarette-only users (26.2% vs. 11.8%, p < 0.001), these results across 10 different physiologic functions cannot be extrapolated specifically to sleep quality based on the study design.

Article 11:

Hopkins, B., Frequency of cigarette and/or e-cigarette use and its associations with sleep health among Canadian adolescents.

The population of this study was adolescents living in Canada. Our population was adults >18 years old located mainly throughout the US with some international participants. This study used unvalidated questions to assess sleep duration only. No validated questionnaire was utilized. Our study assesses sleep quality using the validated PSQI questionnaire. Hopkins, B. found that nicotine users report fewer minutes of sleep per night than non-users, and that high frequency e-cigarette only and e-cigarette dual using adolescents are the least likely to meet the national sleep duration recommendation of 8-10 hours per night compared to non-users. These findings do not address sleep quality.

Article 12:

Kwon, M., Park, E., & Dickerson, S. S. (2019). Adolescent substance use and its association to sleep disturbances: A systematic review.

This is a systematic review of existing literature relating to adolescent substance use and sleep. It is not a primary research article. Its patient population is entirely adolescents where ours is adults >18 years. The writers of this review did conclude that there is evidence for associations between substance use and sleep disturbances in Regularity, Timing, Efficiency, and Duration domains which may result in sleep deprivation, which poses a serious health risk among growing adolescents.

We added the final summary of this literature review to the manuscript, page 2; “Although there have been several studies published over the last few years attempting to address whether e-cigarettes or dual use impact sleep health, many of the studies focused on particular age groups or did not use validated questionnaires for evaluating sleep quality. Others only looked at one type of inhalant rather than all four groups (reference here for which ones only used one group). Of the twelve articles reviewed, one only alluded to a possible relationship between e-cigarette use, oxidative stress, and adverse effects on sleep. Only one study did not find a difference in sleep duration or quality in e-cigarette users, however, this study design had concerning limitations using unvalidated methods for assessing sleep. The remaining ten articles had findings supportive of a relationship between e-cigarette use and decreased sleep quality including sleep duration, onset, latency or regularity. In summary, although the articles all had unique limitations from study design to generalizability of findings, the vast majority of the articles concluded that e-cigarette or dual use leads to alterations in sleep that need further investigation.”

  1. Reviewer Comment: Even after the authors' revision, I maintain that I do not find this study to contribute any scientific knowledge in the relationship between smoking e-cigarette and sleep quality. The analyses are not appropriately conducted to shed light to what the authors claimed, "that the adverse effects of historical or active use of nicotine-based inhalants may be amplified as the COVID-19 pandemic progressed"
  2. Author’s Response:

We apologize that the reviewer does not believe that our study has the ability to contribute to the sparse data existing in this research niche. We do believe that because sleep is known to be negatively impacted during times of increased stress, that our hypothesis that historical or active use of nicotine-based inhalants would have amplified negative correlations with sleep quality as the COVID-19 pandemic progressed was a reasonable one which led to the work that we conducted.

We appreciate the reviewer’s concern. We have extensively reviewed the existing literature and find the data to be quite sparse. Thus, we are unclear why the reviewer believes we have not added any scientific knowledge. We and others have reported improvement in sleep quality during the pandemic, but clearly this finding is quite variable. For example, in the clinic we have seen many patients complain of insomnia which was brought on by the stress of the pandemic. Thus, we believe it is imperative to investigate sources of variability particularly if we can identify reversible causes of disrupted sleep during the pandemic. We hope that our research encourages interventional studies to address smoking and vaping behaviors with a goal of improving overall health including sleep quality. 

Reviewer 2 Report

The article is improved but needs some more statistical analysis. I suggest building multivariable regression model(s) to better explore the relationship between the outcome and the independent variable adjusted for confounders. 

Author Response

Review comment: The article is improved but needs some more statistical analysis. I suggest building multivariable regression model(s) to better explore the relationship between the outcome and the independent variable adjusted for confounders. 

Author’s Response: Thank you for the suggestion, we have run this analysis and added it to our findings. The results of this analysis are included in the results section on page 3 line 152: “Even when controlling for age, gender and presence of any lung disease, inhalant use significantly was associated with a higher PSQI score. PSQI was lower the older you are. Higher in people with lung disease. Higher in inhalant users.  Using multivariate regression, inhalant use, age and lung disease were all independent predictors of the final PSQI score. The parameter estimate for age is -0.030 per year (CI -0.008 to -0.053; p 0.0087), lung disease is 1.040 if present (CI 0.029 to 2.051; 0.0439), inhalant use is 0.810 (CI 0.087 to 1.532; p 0.0283).”

Reviewer 3 Report

Important details have been added in the reviewed version, in particular figure 2, locations of survey participants and figure 3 Responses to the COVID-related questions that were added to the June retake survey.

Considering the multiple locations of study participants, throughout USA and the entire world, it was not possible for the authors to give details on the characteristics of the pandemy, severity  (incidence of new cases, number of hospitalized patients), evolution (improvement or worsening) in each location. However, this is a very important issue which should be indicated in the limitations of the study

Author Response

Reviewer comment: Important details have been added in the reviewed version, in particular figure 2, locations of survey participants and figure 3 Responses to the COVID-related questions that were added to the June retake survey.

Author’s Response: We found your initial feedback regarding our portrayal of this data very helpful and are glad to hear that these additions were well-received. We are grateful for these guiding remarks that allowed us to improve our manuscript.

Reviewer comment: Considering the multiple locations of study participants, throughout USA and the entire world, it was not possible for the authors to give details on the characteristics of the pandemic, severity (incidence of new cases, number of hospitalized patients), evolution (improvement or worsening) in each location. However, this is a very important issue which should be indicated in the limitations of the study

Author’s Response: We agree with the reviewer and appreciate their recommendation. We have added the following the sentence to the discussion: Page 8, “Because of the multiple locations of study participants throughout the US and worldwide, it was not possible to define characteristics of the pandemic including severity and evolution, in each location.”